# Single-Circulating Tumor Cell Whole Genome Amplification to Unravel Cancer Heterogeneity and Actionable Biomarkers

**DOI:** 10.3390/ijms23158386

**Published:** 2022-07-29

**Authors:** Tanzila Khan, Therese M. Becker, Joseph W. Po, Wei Chua, Yafeng Ma

**Affiliations:** 1School of Medicine, Western Sydney University, Campbelltown, NSW 2560, Australia; 19153083@student.westernsydney.edu.au (T.K.); therese.becker@inghaminstitute.org.au (T.M.B.); wei.chua@health.nsw.gov.au (W.C.); 2Medical Oncology, Ingham Institute of Applied Medical Research, Liverpool, NSW 2170, Australia; 3Centre of Circulating Tumor Cells Diagnostics & Research, Ingham Institute of Applied Medical Research, Liverpool, NSW 2170, Australia; joseph.po@hotmail.com; 4South West Sydney Clinical School, University of New South Wales, Liverpool, NSW 2170, Australia; 5Surgical Innovations Unit, Westmead Hospital, Westmead, NSW 2145, Australia; 6Medical Oncology, Liverpool Hospital, Liverpool, NSW 2170, Australia

**Keywords:** whole genome amplification, circulating tumor cell (CTC), single-cell analysis, liquid biopsy, cancer biomarker

## Abstract

The field of single-cell analysis has advanced rapidly in the last decade and is providing new insights into the characterization of intercellular genetic heterogeneity and complexity, especially in human cancer. In this regard, analyzing single circulating tumor cells (CTCs) is becoming particularly attractive due to the easy access to CTCs from simple blood samples called “liquid biopsies”. Analysis of multiple single CTCs has the potential to allow the identification and characterization of cancer heterogeneity to guide best therapy and predict therapeutic response. However, single-CTC analysis is restricted by the low amounts of DNA in a single cell genome. Whole genome amplification (WGA) techniques have emerged as a key step, enabling single-cell downstream molecular analysis. Here, we provide an overview of recent advances in WGA and their applications in the genetic analysis of single CTCs, along with prospective views towards clinical applications. First, we focus on the technical challenges of isolating and recovering single CTCs and then explore different WGA methodologies and recent developments which have been utilized to amplify single cell genomes for further downstream analysis. Lastly, we list a portfolio of CTC studies which employ WGA and single-cell analysis for genetic heterogeneity and biomarker detection.

## 1. Introduction

Circulating and disseminated tumor cells (CTCs and DTCs) are cancer cells that dissociate from primary and metastatic cancer sites and enter the circulation with potential to seed distant metastases. CTCs can be enriched or isolated from a simple blood liquid biopsy. Such biopsies are economic and repeatable, allowing the monitoring of changes in cancer longitudinally. Moreover, it is broadly recognized that cancers are heterogeneous [1]; liquid biopsies enable sampling of CTCs released from many tumor sites in comparison to a small, isolated tissue sample, which would not be representative of the entire tumor burden. Heterogeneity may develop early during cancer development. In general, alterations in tumor suppressors or oncogenes predispose cells to cancerous transformation, and such genes, often referred to as “driver genes”, grant mutated cells growth and survival advantages [1]. Due to common genetic instability, cancer cells further acquire and accumulate genomic changes, such as copy number variations (CNVs) or other genetic mutations, over time which can pass on to “daughter” cells. Often, cells at this stage carry their own unique mutation signatures and thus generate heterogeneity. Some of these changes may prove to be beneficial in resistance to therapeutic drugs. Indeed, it has been shown that cancers with a high tumor mutation burden (the number of different genetic changes acquired) are more likely to be treatment-resistant but respond to immue checkpoint inhibitor therapy [2]. For example, higher mutational load (top 20% highest TMB quintile) predicts better survival across diverse cancer types tested, except for glioma, and this is relevant in patients treated with either anti-PD-1 (programmed cell death protein 1) or anti-CTLA-4 (cytotoxic T lymphocyte-associated antigen-4) therapies [3]. Thus, single-CTC/DTC analysis provides a strategy to characterize cancer heterogeneity, cancer evolution and metastatic potential, as well as to predict and monitor developing resistance and guide best personalized therapy [4,5]. However, an intrinsic property of CTCs is their rarity, posing technical challenges for the isolation of CTCs for single-cell analysis [6]. The entire genomic content of a normal human cell is about 6–10 pg [7], which is insufficient for most genomic analyses, as a larger quantity of DNA is essential to generate high-quality libraries for sequencing or retain reliable data for most mutation detection assays. Therefore, whole genome amplification (WGA) has become a prerequisite to obtain sufficient genetic material from single CTC to perform genetic analysis [8]. In this review, we evaluate WGA- and CTC-related studies published in the last decade (2012–2022) with a focus on the recent advances in CTC isolation and recovery and the technical advances in WGA, and we further analyze studies on the main cancer types that combined CTC analysis with WGA.

## 2. CTC, WGA and Single-Cell Analysis

### 2.1. CTC Isolation, Identification and Recovery

CTCs, sourced from liquid biopsy, are a proven and potent prognostic biomarker in multiple metastatic cancer types and have been investigated as treatment-outcome parameters in phase I and II clinical trials (reviewed in [9]). However, CTCs are rare (1–10 in 10^6^ lymphocytes), with high turn-over dynamics [10]. Strategies to efficiently isolate or enrich CTCs while not biasing for specific subpopulations are technically challenging but critically required for downstream single-CTC analysis (Figure 1). Most CTC isolation technologies utilize enrichment techniques based on tumor cell physical and biological properties, including size, deformability, surface charge, density and cell-surface expressing markers [9]. For example, CellSearch™ (the only FDA-approved platform for the enumeration of breast, prostate, and colorectal CTCs) and MACS™ (magnetic activated cell sorting) are immunomagnetic positive enrichment-based methods [11,12,13]. Technologies such as CTC-Chip™ and Isoflux™ combine microfluidics with positive immunocapture, which can be customized to incorporate additional cancer markers targeting epithelial, mesenchymal, cancer type-specific or stem cell markers, such as EpCAM (epithelial cell adhesion molecule), EGFR (epidermal growth factor receptor), vimentin and CD44 [14,15]. In some cases, antibody cocktails targeting several cancer cell-surface proteins simultaneously result in better coverage of CTC subpopulations [16]. Membrane filtration size-based methods (ISET™ and ScreenCell™) [6,17] are cost-effective but associated with technical issues, such as blocked membranes. Additionally, not all CTCs are different in size to blood cells and filtration enrichment would therefore exclude small CTCs. There are alternative CTC enrichment methods, including microfluidic size and deformability-based methods (Parsortix™), and negative selection-based methods (RosetteSep™ and EasySep™). However, the above-mentioned methods are only applicable to small blood volumes in vitro. An innovative device CellCollector™ was developed recently to overcome this limitation and isolate CTCs in vivo, with more frequent discovery of CTCs. Generally, CTC isolation platforms require special expertise and instruments.

Virtually all CTC enrichment methods, unless based on single cell separation, are still yielding samples with a high background of residual lymphocytes, which necessitates identifying and recovering CTCs for single-cell analysis. CTC identification is achieved with staining of cancer cell-specific markers, such as pan-cytokeratin or cancer type-specific markers, e.g., PSMA (prostate-specific membrane antigen) for prostate cancer CTCs [6,9]. Furthermore, CTC identification by probing several cancer-specific markers is advantageous in certain cancer types with higher heterogeneity, as reported for melanoma CTCs [18]. Single CTCs can be recovered by micromanipulations [19,20,21], serial dilutions, flow cytometry sorting [22,23] or capture of CTCs in wells [6], as well as by recent DEPArray technology that combines microfluidics and electric fields based on dielectrophoresis (DEP) [24,25,26,27].

### 2.2. Challenges and Technical Advances of WGA

Three main WGA principles have been developed and are widely used: degenerate oligonucleotide-primed PCR (DOP-PCR), multiple displacement amplification (MDA) and multiple annealing and looping-based amplification cycles (MALBAC) (Figure 2A–C). Application of these WGA technologies are now commercially available in kit form from various suppliers (Table 1). It is noteworthy that each technology has its own advantages and disadvantages in terms of coverage and bias, and this may be relevant depending on the planned downstream assays. The suitability or compatibility of each method needs careful validation, and several comparative studies have been recently published to compare such WGA kits in the context of their applications for different downstream analyses [28,29,30,31,32]. The main concern with WGA is how to best amplify the single copy of a genome while minimizing the introduction of sequence loss and technical artefacts (amplification bias, genome loss, mutations and chimeras, false positivity, and negativity). DOP-PCR is an exponential amplification with random priming which results in over- or under-amplification of genomic areas and ultimately low genome coverage due to amplification bias (Figure 2A) [33]. MDA works under isothermal conditions and offers exponential amplification, thus potentially causing sequence-dependent bias. It is most widely used; however, it is a non-uniform amplification, not reproducible from cell to cell (Figure 2B) [33]. Recently, considerable effort has been made to improve MDA methods in terms of scalability, improved uniformity and coverage, along with reduced contamination, leading to the development of microchannel MDA [34], single-droplet MDA [35,36] and centrifugal-driven droplet MDA [37,38]. While unlike MDA, MALBAC provides quasi-linear amplification, which reduces sequence-dependent bias and results in better CNV detection accuracy, with a low rate of false-negative results [33]. The preamplification is followed by exponential PCR amplification, producing DNA fragments required for NGS (Figure 2C). MDA and MALBAC have comparable levels of genome coverage that are significantly higher than that of DOP-PCR. In terms of uniformity, DOP-PCR gives the flattest CNV raw data, without any data normalization. Allelic dropout (ADO) is related to the presence of SNVs (single nucleotide variants) situated inside the primer sequences. ADO is the limiting factor in the detection of mutations in heterozygous samples, which is lower in MALBAC as compared to DOP-PCR or MDA [33].

Recently, a novel single-cell WGA method, linear amplification via transposon insertion (LIANTI) (Figure 2D), has been established which combines Tn5 transposition and T7 in vitro transcription. It eliminates non-specific priming and exponential amplification. LIANTI demonstrates 97% genome coverage and 17% ADO and has the lowest false-positive rate for SNV detection compared with previous methods [39]. Another novel isothermal WGA method is based on primary template-directed amplification (PTA) (Figure 2E), which has been demonstrated to be more uniform and accurate than other exisiting approaches, with significant improved variant calling sensitivity and precision; the authors further developed a tool to map genome-wide interactions of mutagens at base-pair resolution [40]. Further, META-CS (multiplexed end-tagging amplification of complementary strands) (Figure 2F) has been proposed to largely reduce false positives and improve the accurancy of SNV detection by combining improved Tn5 transposition and pre-amplification of the complementary strands of double-stranded DNA [41]. Recently, WGA employing water-in-oil emulsion after single cell lysis and addition of MDA reaction mix to generate picoliter droplets, as used for droplet digital PCR (ddPCR), has been established to assure more even amplification throughout the genome [38]. Instead of using microfluidic chips or spinnning capillaries with oil, which need special instruments and training, Fu et al. combined micro-capillary array-based centrifugal droplet generation with emulsion MDA, which is highly scalable to 48 samples in a single centrifugal run. Notably, a novel “whole blood in, WGA product out” microfluidic chip was developed recently to perform blood filtering, CTC enrichment and isolation, lysis and WGA at single cell level in a single chip, minimizing cell loss and potential contaminations [42]. These and other new emerging technologies are bound to revolutionize the utility of WGA to deliver more precise means for downstream analysis.

In the CTC research field, due to the long duration of work pipelines, CTC fixation prior to WGA is preferable. However, cell fixation crosslinks DNA and histones, thereby reducing chromatin accessibility, introducing some amplification bias. Different fixatives can cause differential sequence quality, yield and ADO rates [43]. In addition, the inconsistent quality of WGA products is an ongoing issue, and it is advised to perform quality-control (QC) tests before progressing samples to costly downstream analyses [44].

### 2.3. Downstream Analysis of Single-Cell WGAs

First generation DNA “Sanger” sequencing, next-generation sequcing (NGS) and array comparative genomic hybridization (aCGH) are common downstream technologies used to discover the genetic profiles of single cells following WGA. Sanger sequencing is the most widely used method for SNVs, point mutations, deletions/duplications and mosaic mutation detections [45]. NGS refers to sequencing of the whole genome or exome; alternatively, it may target selected genes, usually the most relevant oncogenes/tumor suppressor genes, which may be “oncomined” in gene panels available broadly or for specific cancers. Illumina and Thermo Fisher Scientific support the most popular NGS platforms [46]. aCGH is the method of choice for genome-wide detection of CNVs [47]. PCR-based assays, such as ddPCR, are also frequently applied to screen for known variants. Strategic experimental designs, data preprocessing, filtering and normalisation and readmapping are all crucial for good genetic profiling [48]. These fields are also developing rapidly, and although tracing these developments is beyond the scope of this review, they have been reviewed previously [49,50,51].

**Figure 2 ijms-23-08386-f002:**
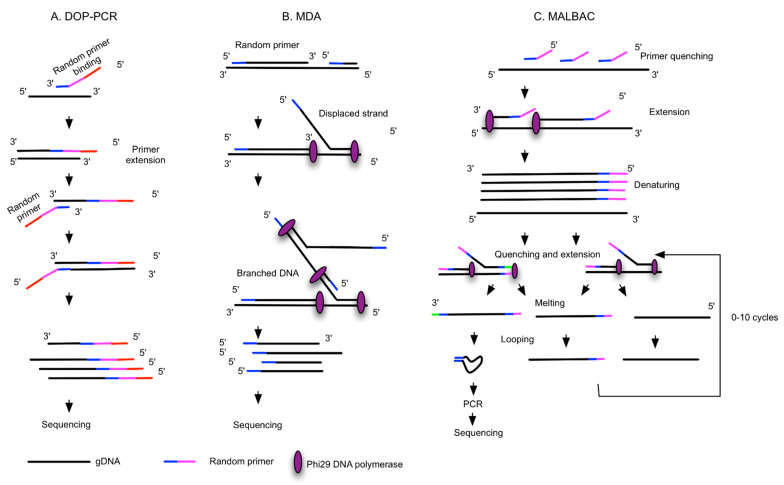
Different WGA principles. The schematic workflows are simplified and adapted from [33,39,41]. (**A**) Degenerate oligonucleotide-primed PCR (DOP-PCR): random priming with degenerate oligonucleotide primers (DOPs) and PCR. (**B**) Multiple displacement amplification (MDA): random priming and isothermal amplification with phi29 DNA polymerase, with strong strand displacement activity. (**C**) Multiple annealing and looping-based amplification cycles (MALBAC) involve random primers with fixed sequences for the amplification of linearly original gDNA to form semi-amplicons, and full amplicons are further amplified and form DNA loops attributable to the complementary sequences at 5′ and 3′ ends. DNA loops are PCR-amplified. (**D**) Linear amplification via transposon insertion (LIANTI) uses gDNA fragmented by the Tn5 transposome and tagged with sequences containing the T7 promoter. T7 RNA polymerase binds the promoter and linearly amplified RNA, and cDNA is generated by reverse transcription and further tagged with barcodes for sequencing. (**E**) Primary template-directed amplification (PTA) utilizes phi29 polymerase and exonuclease-resistant terminators to create small double-stranded amplicons that undergo limited quasilinear processes, with more amplifications occurring based on the primary template. (**F**) Multiplexed end-tagging amplification of complementary strands (META-CS) works via transposome complexes that form from a 1:1 molar ratio of Tn5 transposase and a mixture of 16 unique transposons, which allows DNA fragmentation and tagging with two random transposon sequences. Forward and reverse strands of original DNA are pre-amplified to obtain strand-specific labelling.

**Table 1 ijms-23-08386-t001:** WGA methods.

WGA Method	Principles and Polymerase	Commercial Kits	Technical Challenges	Advantages	Preferred Downstream Analysis
DOP-PCR [33,50]	Random priming and PCR amplification, Taq Polymerases	Sigma GenomePlex Single Cell WGA Kit, PerkinElmer DOPlify WGA kit	Low genome coverage (40–50%), better uniformity of amplification, high FP and FN, low success rate	Quick, no need of normalization	CNV, STR analysis
MDA and improved MDA [52,53]	Random priming and isothermal exponential amplification, Phi29 or *Bst* polymerases	Qiagen REPLI-g Single Cell Kit, GE GenomiPhi DNA Amplification Kit, AmpliQ Genomic Amplifier Kit, Sygnis TruePrime WGA kit	Less uniformity, artifact of C>T transitional mutation, non-reproducible from cell to cell, low chimera rate	More genome coverage (80%), low FP and FN, compatible with digital droplet MDA	Mutation detection, SNP
MALBAC [52,54,55,56,57]	Isothermal preamplification and PCR, *Bst* polymerase, deep vent (exo-) DNA Polymerase	Yikon Genomics Single Cell WGA Kit, Rubicon Genomics PicoPLEX WGA Kit, TakaRa PicoPLEX	Complicated procedure, intermediate coverage and uniformity, intermediate FP and FN	Reproducible from cell to cell, low ADO	CNV
LIANTI [39]	Random fragments tagged by T7 promoters, linear amplification of RNA, reverse transcription	NA	Needs further study	High genome coverage (97%) and low ADO (17%), low FP for SNV detection	SNV
PTA [40]	Isothermal WGA, quasi-linear process, Phi29 polymerase	BioSkryb ResolveDNA WGA kit	Needs further study	High coverage (95%), reproducible, high uniformity and accuracy, compatible with high-throughput reactions in microfluidic devices or emulsions	Improved capacity to call SNVs, CNVs and SVs; superior SNV sensitivity
META-CS [41]	Fragmented by Tn5 transposase, randomly tagged with transposon sequences, DNA pre-amplification	NA	Needs further study	High success rate (90%), single tube reaction to minimize loss, high amplification uniformity	SNVs, insertions, deletions, SVs

Note: ADO: allelic dropout; CNV: copy number variant; DOP-PCR: degenerate oligonucleotide primed PCR; FP: false positive; FN: false negative; LIANTI: linear amplification via transposon insertion; MALBAC: multiple annealing and looping-based amplification cycles; MDA: multiple displacement amplification; META-CS: multiplexed end-tagging amplification of complementary strands; STR: short tandem repeat; SNV: single nucleotide variant; SNP: single nucleotide polymorphism; SV: structural variant; NA: not available.

## 3. Single-Cell Analysis of CTCs and Biomarker Detections

In this section, CTC studies for various cancer types are collated and a summary of CTC isolation and WGA methods, along with the relevant main findings, is presented in Table 2. The main studies are further discussed in the following subsections.

### 3.1. Breast Cancer

Breast cancer (BC) is the most common female cancer and CTC is a predictive marker of poor survival and metastatic relapse [58]. The detection rate of CTCs correlates with the number of metastatic sites, and BC patients with brain metastasis may have the highest CTC counts [59].

The hormone status of BC, such as expression of the estrogen receptor (ER) or progesterone receptor (PR), indicates the feasibility of ER-targeted endocrine therapy [60]. However, no correlation was found between total CTC number and/or ER expression status as determined by immunocytostaining and the intensity of ER staining in primary tumors [20]. Only 81.3% of patients were positive for ER expression in CTCs, while ER-negative CTCs were also found in ER-positive patients, delineating the genetic inconsistencies between CTC counts. ER status in CTCs might have predictive power with regard to response and resistance to endocrine therapy and may thus help in the choice of better treatment options [20]. One study performed Sanger sequencing on CTC WGAs (MALBAC), which resulted in the identification of the *ESR1*-Y537S variant known to produce a constitutively active receptor and *ESR1*-T570I (a novel mutation) in exon 8 [25]. This study found *ESR1*-Y537S heterozygously and homozygously in single CTCs and confirmed mutations in matched cell-free DNA (cfDNA) in one patient. Interestingly, in another patient, heterozygote *ESR1*-T570I and homozygote *ESR1*-Y537S were found in a single CTC, but *ESR1*-T570I could not be detected in matched cfDNA [25]. Thus, using two entities extractable from a blood biopsy, CTCs and cfDNA biomarkers may complement each other and enhance the chance of finding disease-related variants. However, in another study that screened for exon 4, 6 and 8 *ESR1* mutations after WGA (Picoplex, MALBAC), none was found in individual CTCs [20].

The PI3K/AKT/mTOR pathway (Phosphoinositide 3-kinase/protein kinase B/mammalian target of rapamycin) regulates cell growth, survival, and angiogenesis. Upregulated activity has been linked to oncogenesis and is a major therapeutic target [61]. In BC, mutations in PIK3CA are found in about 40% of ER-positive cancers and have been implicated in resistance to HER2-based therapies [62]. Pharmacologic targeting of PIK3Ca in HR (hormone receptor) +/HER2-metastatic BC offers significant benefits to patients with endocrine therapy resistance [44]. Several single CTC-based studies [19,26,27,44,63] were conducted to study mutations in the PIK3CA gene. Heterogenous expression of PIK3CA mutations among CTCs and matched primary tumors, and even among CTCs from the same patient, was observed. Individual PIK3CA mutations found in Ampli1-amplified CTCs included E542K and H1047R [44], as well as E542K, E545K and H1047R, as was determined in a second study [26]. Another study found PIK3CA mutations (E542K, E545K, H1047R, H1047L and M1043V) in exon 9 and 20 in at least one CTC in 36.4% of the patients tested [64]; similar data were reported in other studies [27,63] (Table 2). Neumann et al. analyzed CTCs from two patients and mutations (SNP, G > A, E545K) in PIK3CA were confirmed in CTCs from one patient but not in any of the CTCs from the second patient [65]. All these studies were conducted using Ampli1-based CTC WGA. Importantly, mutations in PIK3CA have been linked to resistance to receptor tyrosine-protein kinase HER2 (Erb-B2, ERBB2)-targeted therapies [66,67]. *ERBB2* amplification was detectable by qPCR in 10.9% of single CTCs after WGA but was not detected in WGA samples of single white blood cells (WBCs). *ERBB2* copy numbers as determined by aCGH matched the qPCR results, with only two samples showing conflicting results [27].

Similarly, WGA (Ampli1) of single BC CTCs allowed mutation detection in *TP53*, a critical tumor suppressor that is associated with poor diagnosis, drug resistance, increased proliferation, and invasion in BC [68]. In contrast, another study did not find any *TP53* mutations in BC CTCs after WGA (Ampli1) [63]. Cyclin D1 is a central cell-cycle regulator, and amplification of the gene encoding cyclin D1, *CCND1*, has been shown to be associated with BC overall survival [69]. WGA (Ampli1) of CTCs enabled aCGH detection of *CCND1* amplification in 46.2% of 26 CTCs from 13 BC patients [63].

Approximately 40–50% of metastatic BC patients are diagnosed with liver metastasis, which is associated with certain CNVs, including β-defensin (hBDs) and defensin genes, which are implicated in anti-angiogenesis and immunomodulation signaling pathways [21]. The presence of CTCs was associated with recurrence and shorter disease-free survival in BC patients with liver metastases. The CNV patterns detected in WGA (MALBAC) of CTCs were comparable to those observed in freshly diagnosed liver metastasis but different to those in recurrent liver metastasis, warranting further analysis [21].

Finally, Wang et al. reported shared SNVs between CTC WGAs and primary tumors, indicating clonality and the origination of CTCs from primary tumors. The sequencing results defined 22 co-occurring mutated genes shared between CTCs and matched primary tumors, and, interestingly, 394 SNVs were shared by at least two CTCs. Common mutations affecting *LRP1B,* and *APC* were co-occurring between bulk tissue and CTC shared SNVs. This type of analysis can shed light on tumor development, heterogeneity and ultimately may be linked to therapeutic options [22]. 

**Table 2 ijms-23-08386-t002:** The application of WGA and biomarker detection of single CTCs in various cancer types.

Studies (Author, Year)	CTC Isolation	CTC Recovery	WGA Kits	Downstream Molecular Analysis	CTCs+ Patients Analyzed	CTC Nr Analyzed for WGA	Main Findings in Genetic Mutations and Alterations
*mBC or HER2- mBC*							
Babayan, A. et al., 2013 [20]	Density gradient	Micromanipulator TransferMan NK2	PicoPlex	Multiplex PCR	4	8 single CTCs	*ESR1* mutations in exons 4, 6 and 8 were not found
De Luca, F. et al., 2016 [68]	CellSearch	DEPArray	Ampli1	NGS (Ion AmpliSeq Cancer Hotspot panel v2)	4	3–5 single CTCs per patient	51 sequence variants in 25 genes were found, including somatic mutations in *TP53* (8 mutations) and *PDGFRA* (3 mutations). High intra- and inter-patient heterogeneity, discordance in mutational status between CTCs and primary tissue
Gasch, C. et al., 2016 [64]	CellSearch	Micromanipulator TransferMan NK2	GenomiPhi, Ampli1	Sanger sequencing, PCR	33	114 single CTCs	*PIK3CA* mutations in exon 9 and 20
Kaur, P. et al., 2020 [70]	Microfluidic ANGLE Parsortix	NA	REPLI-g	WES (SNVs, CNAs and SVs)	5	5 CTCs and 5 WBCs	Elevated C>T mutational signature in patient samples. Low VAFs for somatic variants in CTCs compared to metastasis, complex rearrangement patterns were observed, high discordance between paired samples, marked heterogeneity of somatic landscape
Li, S. et al., 2020 [59]	CellCollector	CellCollector	REPLI-g	NGS (HiSeq X-Ten Illumina)	17	0–15 CTCs	Different metastatic sites have their own corresponding high-frequency mutation genes
Neumann, M. H. et al., 2016 [65]	CellSearch	CellCelector	Ampli1	For library preparation, the multiplex PCR-based Ion Torrent AmpliSeqTM technology with Ampli1 CHPCustom Beta panel	2	7 single CTCs	Functional *PIK3CA* SNP (G to A, E545K) was detected in CTCs of patient 1 but not in CTCs of patient 2
Neves, R. P. et al., 2014 [63]	CellSearch	FACS	Ampli1	aCGH (CNAs), qPCR	30	192 single CTCs	72.9% WGA success rate, 46.2% of WGA products show *CCND1* amplification, mutations in *PIK3CA* exon 20 in c.3140 were found in CTCs (2/12 analyzed patients), *TP53* mutations in exons 5, 7 and 8 were not found
Paolillo, C. et al., 2017 [25]	CellSearch	DEPArray	MALBAC	Sanger sequencing	3	40 single CTCs and 12 WBCs	*ESR1* mutations (Y537S and T570I) were identified
Pestrin, M. et al., 2014 [26]	CellSearch	DEPArray	Ampli1	Sanger sequencing (hotspot regions in PIK3CA exon 9, 20)	18	115 single CTCs	33% of patients had an identified *PI3KCA* mutation. Six different mutations in the *PI3KCA* gene, such as c.3140A>G, c.1633G>A, c.1624G>A, c.1624G>A, etc., were identified
Polzer, B. et al., 2014 [27]	CellSearch	DEPArray	Ampli1	ERBB2 qPCR (CNV), PIK3CA Sequencing, aCGH	66	510 single CTCs and 189 leukocytes	*PIK3CA* mutations in exon 9 and 20. Analysis of ERBB2 alterations
Schneck, H. et al., 2013 [44]	CellSearch	NA	Ampli1	Multiplex PCR, SNaPshot	44	NA	*PIK3CA* mutations in exon 9 and 20, such as E545K and H1047R, were detected, but E542K, E545G and E545A were not found
Wang, Y. et al., 2018 [22]	FACS combined with oHSV1-hTERT-GFP viral infection	FACS	MALBAC	WGS for CTC, WGS and WES for matched primary and metastatic tissue	8	11 single CTCs	SNVs accumulated sporadically among CTCs and matched primary tumors, at least 2 CTCs shared 394 SNVs, SNV mutations in *APC* and *LRP1B* genes co-occurred in CTC-shared and bulk tissue, CTC behaviour-related SNVs were verified
Zou, L. et al., 2020 [21]	CellSearch	Micropipetting	MALBAC	WGS (CNV and gene set enrichment analysis)	2	Single CTCs, but number is unknown	Different frequencies of CNVs between newly diagnosed and recurrent liver metastasis; similar CNV patterns among isolated CTCs of recurrent BCLM and recurrent liver metastasis; 25 genes were identified as CNV signatures of BCLM, including β-defensins and defensins
*PC or mCRPC*							
Faugeroux, V. et al., 2018 [6]	ISET filtration, CellSearch, Rosettesep	Self-seeding microwell chips, FACS, laser microdissection	Ampli1	WES (10x depth coverage)	11	179 WGA samples or 34 WES	Shared *GRM8*, *TP53* and *PTEN* mutations in epithelial CTC samples and other CTC-exclusive variants
Greene, S. B. et al., 2016 [71]	Epic Sciences	Eppendorf TransferMan NK4 micromanipulator	SeqPlex Enhanced	Sequencing with Illumina NextSeq500 using a High Output kit in a Paired-End 2x150 format (PE 2x150) (CNV)	7	67 single CTCs	*AR* amplification and *PTEN* loss
Gupta, S. et al., 2016 [72]	CellSearch, RBC lysis and CD45 depletion	IE/FACS	RepliGene, WGA4	aCGH (CNV)	16	16 CTCs and matched leukocytes	*AR* amplification in 50% of CTC samples, *ERG* genomic amplification in 40% of patients, *PTEN* loss, genomic alteration in chromatin reading and proliferative pathways
Magbanua, M. J. et al., 2012 [73]	CellSearch, IE/FACS	IE/FACS	WGA4	aCGH	12	9 patient bulk CTCs	Gains in 8q and loss in 8p; gains in the *AR* region of chr X of CTCs, including *AR* gains in 78% of cases
Rangel-Pozzo, A. et al., 2020 [17]	ScreenCell filtration	Laser microdissection	Ampli1	WES	9	21 single CTCs and 4 lymphocytes	Genetic variations in nine telomere maintenance pathways, including telomeric repeat-binding factor 2 (TRF2), SNVs and indels associated with telomere maintenance genes and known cancer drug response; presence of CNAs in 11 different pathways, including the DNA damage repair (DDR) pathway
Wu, Y. et al., 2016 [4]	Density gradient, negative and positive selection with magnetic beads	Laser microdissection	PicoPLEX (<40 cells), WGA2 kit (GenomePlex for microdissected tissues)	SNP array profiling (CytoSNP-12 and omni1-Quad bead chips, NspI 250k, SNP6.0, and CytoScanHD arrays), Nanostring (nCounter Cancer CN panel)	8	8 disseminated tumor cells (bulk cells)	Gain of Ch 7 and 8q, loss in 8p, 12q23, 10q26, 13q and 16q21. *AR* gain, *TMPRSS2/ERG* alterations and *MYC* and other gained regions, *FOXO1* gene deletion
*Lung Cancer*							
He, Y. et al., 2017 [74]	CellCollector	CellCollector	REPLI-g	NGS (hotspot panel v2)	5	6 CTCs	44 cancer-related genes existed in mutations in the analyzed CTCs and some cancer-related mutations were identified in *KIT*, *SMARCB1* and *TP53* genes
Lu, S. et al., 2020 [28]	CellSearch	DEPArray	MALBAC, REPLI-g, WGA4, Ampli1	Targeted sequencing, WES, WGS	4	80 single CTCs and 11 WBCs	Comparative study, MALBAC WGA coupled with LP-WGS is a robust workflow for CNV profiling, but none of the WGA methods achieve sufficient sensitivity and specificity by WES
Mariscal, J. et al., 2016 [75]	CELLection Epithelial Enrich Dynabeads	NA	WTA2	Gene expression profiling (Agilent 4x44k gene expression arrays), qPCR	42 NSCLC patients and 16 controls	NA	CTC-specific expression profile associates with the PI3K/AKT, ERK1/2 and NF-kB pathways. *NOTCH1*, *PTP4A3*, *LGALS3* and *ITGB3* were further validated by RT-qPCR in an independent cohort of NSCLC patients
Nakamura, I. T. et al., 2021 [13]	AutoMACS	DEPArray	SMARTer PicoPLEX	NGS (Todai OncoPanel, AmpliSeq for Illumina comprehensive cancer panel, WGS) and Sanger sequencing	2	40 single floating tumor cells in pleural effusion	*EGFR* exon 19 deletion was confirmed in 63.2% of samples from case 1, detection of 85% *EML4-ALK* fusion in case 2, alectinib- resistant mutation of *ALK* (p.G1202R) in case 2. A *BRCA1* truncating mutation and an *RAF1* oncogenic mutation were identified
Ni, X. et al., 2013 [5]	CellSearch	Micropipetting	MALBAC	WGS at ∼0.1× sequencing depth and WES for SNV/indel	11	72 single CTCs (including 4 leucocytes)	*EGFR* mutations (such as one INDEL p.K746_A750del), *PIK3CA* (such as p.E545K), *RB1* (p.R320*) and *TP53* mutations (such as p.T155I) were only shared between the liver metastatic tumor and CTCs; gain region in chromosome 8q contains the c-Myc gene; gain in chromosome 5p, which contains the telomerase reverse transcriptase (*TERT*) gene; chromosomal regions, including 3q29, 17q22, 17q25.3 and 20p13, had significant gain in all 19 CTCs of patients
*Colorectal Cancer*							
Fabbri, F. et al., 2013 [76]	OncoQuick	DEPArray	Ampli1	Sequencing and pyrosequencing	21	16 samples or cases	*KRAS* gene mutations in 50% of cases. G12C, G12D and G13D-KRAS mutations in one patient in three different groups of CTCs
Gasch, C. et al., 2013 [19]	CellSearch	Micromanipulator TransferMan NK2	GenomePlex, GenomiPhi	Targeted sequencing for KRAS, BRAF and PIK3CA gene, qPCR for EGFR	5	69 single CTCs	*EGFR* amplification in 7/26 CTCs, *KRAS* mutations (G12V) in 33% of CTCs, *PIK3CA* mutations (E545A and E542K) in 39% of CTCs, no *BRAF* locus change detected
Li, R. et al., 2019 [42]	Microfluidic chip (SCIGA-chip)	Microfluidic chip (SCIGA-chip)	MDA	Illumina sequencing (SNPs/SVs)	1	2 single CTCs and 1 WBC	A novel method involving all processing steps from blood collection to WGA preparation, 11 shared somatic mutations (e.g., *C18orf25*, *GFM2*, *DDX60L*, etc.) and 153 structure variations were identified
*Pancreatic Cancer*							
Court, C.M et al., 2016 [77]	Density gradient and NanoVelcro/LCM microchip	Laser microdissection	REPLI-g	Sanger sequencing	12	119 single CTCs and 103 WBCs	*KRAS* mutations in 92% of patients and 33 out of 119 single CTCs sequenced (resulting in a 27.7% detection rate in single CTCs). No *KRAS* mutants were found in any WBCs
*Melanoma*							
Reid, A. L. et al., 2014 [78]	RBC lysis, immune-magnetic beads	NA	REPLI-g	ddPCR and castPCR	15	30 CTCs	Comparative study of ddPCR and castPCR. *BRAF*-V600E/K mutations were detected
Ruiz, C. et al., 2016 [79]	RBC lysis	Micromanipulator	GenomePlex	CNV analysis	40	Single CTCs and WBCs	Deletions of *CDKN2A* and *PTEN*; amplifications of *BRAF*, *TERT*, *MDM2* and *KRAS*; chromosomal amplifications in chr12, 17 and 19
*Mixed patient cohort*							
Aljohani, H.M. et al., 2018 [23]	RBC lysis, CD45 depletion and EpCam positive selection	FACS	REPLI-g	Sanger sequencing, ddPCR	10	NA	Mutations (R34G, E79Q, E82G) in *Nrf2* in isolated CTCs, some mutations in the Keap/Nrf2/ARE pathway
Ferrarini, A. et al., 2018 [80]	CellSearch	DEPArray	Ampli1	WGS (CNAs), aCGH	3	15 single CTCs and 7 WBCs	A large amplification (100 Mbp) on chr 8, including the *c-MYC* gene, copy number loss was detected in the *BRCA2* locus
Gao, Y. et al., 2017 [81]	CellSearch	Micropipetting	MALBAC	WGS and WES for SNV/indels, SVs, CNs	23	97 single CTCs	Homozygous deletion of *PTEN*; amplification of the *MYC* gene; 11 focal regions were identified, including well-known tumor suppressor genes or oncogenes, which were deleted or amplified

Note: aCGH: array comparative genomic hybridization; chr: chromosome; CNA: copy number alteration; CNV: copy number variant; mCRPC: metastatic castration resistant prostate cancer; ddPCR: droplet digital PCR; FACS: fluorescence activated cell sorting; IE: immunomagnetic enrichment; ddPCR: droplet digital polymerase chain reaction; RBC: red blood cell; SNV: single nucleotide variant; SNP: single nucleotide polymorphism; SV: structural variant; WBC: white blood cell; WES: whole exome sequencing; WGA4 and WGA2: different versions of GenomePlex; WGS: whole genome sequencing; WTA: whole transcriptome amplification; WTS: whole transcriptome sequencing; NA: not available.

### 3.2. Prostate Cancer

Prostate cancer (PC) is the most common cancer type diagnosed in men; eventually, it develops into castrate-resistant prostate cancer (CRPC) following standard of care androgen deprivation therapy (ADT). Commonly altered genes during CRPC progression include *AR* (androgen receptor), *ERG* (ETS-related gene), *c-MET* (tyrosine-protein kinase MET), *PTEN* (phosphatase and tension homology deleted on chromosome 10) and *PI3K/AKT* signaling pathway genes. *AR* alterations in CTCs, especially *AR* amplification and expression of splice variant AR-V7, predict poor treatment outcomes for ADT [71,72,82,83]. *ERG* amplification of CTCs is also informative for treatment selection and might contribute to resistance to taxane therapy [72].

WGA-based single-CTC analysis found significant numbers of shared mutations in *PTEN*, *GRM8* and *TP53* among PC CTCs, particularly if they were of epithelial phenotype. Some recurrent mutations found in CTCs correlated with matched metastatic tissue. Interestingly, sequencing multiple CTCs did not significantly change the number of mutations found [6]. This may indicate that heterogeneity is less of an issue, as these mutations may be shared by most CTCs and are likely early events in cancer formation. Both epithelial and non-epithelial CTCs showed CTC-exclusive alterations affecting invasion, DNA repair mechanism, cancer-driver, and cytoskeleton genes [6]. The shared mutations between matched tissue and CTCs might provide insights into the metastatic spread of cancer and the origins of CTCs, as it is assumed that more mutations are acquired during cancer progression and spread.

aCGH analysis of CTC WGA products from CRPC patients demonstrated genomic gains in >25% of CTCs. Such genomic gains were observed in *AR*, *FOXA1*, *ABL1*, *MET*, *ERG*, *CDK12*, *BRD4* and *ZFHX3*, while common genomic losses involved *PTEN*, *ZFHX3*, *PDE4DIP*, *RAF1* and *GATA2*. *AR* and *NCOA2* amplification were found in 50% and 43.75% of CTC WGAs, respectively, while *ERG* amplification was found in 40% of patient CTCs. Loss of *KDM6A* was found in 6.25%, while *KDM6A* gain was found in 50% of mCRPC CTC samples. *MYCN* gene amplification was observed after the development of enzalutamide resistance. Similarly, *PTEN* gain was observed before starting enzalutamide, and *PTEN* loss appeared after enzalutamide treatment [72]. Another aCGH analysis of WGA CTCs found *AR* gain in 78% of nine patient bulk CTC samples (that is, samples combining more than a single CTC). However, *AR* gain in CTC WGA samples is not always found in matched tissues and may be due to previous archival tissues failing to represent tumor evolution; nevertheless, some copy number alterations, including gains and losses of chromosome 8p and 8q, are concordant between CTCs and primary tumors [73].

### 3.3. Lung Cancer

The detection of certain driver mutations, such as in *EGFR* and *ALK* fusion, is associated with the early stages of lung cancer, its development and drug resistance [74]. Genetic analysis of CTCs from the same patient can give overall information about deletions, fusions, insertions and SNVs in the metastatic tumor and such changes can be monitored during treatment, even in the presence of cell-to-cell heterogeneity; however, a large number of CTCs needs to be sequenced [5].

Ni. et al. observed number of mutations in different genes, such as *EGFR, PIK3CA, RB1* and *TP53*, after exome sequencing of single-CTC WGA products. Amongst these alterations, one INDEL in the *EGFR* gene (K746_A750del), which is a target for tyrosine kinase inhibitors (TKIs), was found in CTCs as well as in the primary and metastatic tumors of the patients, while other mutations in *PIK3CA* (E545K), *TP53* (T155I) and *RB1* (R320*) genes were only observed in CTCs and metastatic tumors in the liver. This study also found some common CNV regions that have important roles in cancer development, such as cell proliferation, differentiation and protecting chromosomal ends from degradation. These regions include regions of gain in chromosome 8q, the *c-Myc* gene loci, and in chromosome 5p, the *TERT* gene (telomerase reverse transcriptase) loci, 17q22, 17q25.3 and 20p13. The CNV patterns of individual CTCs from the same patient were reproducible. It was also found that CNV patterns were not changed upon different drug treatments [5].

Floating tumor cells (FTCs) from the pleural fluid of lung adenocarcinoma patients were enriched and amplified. *EGFR* exon 19 deletion (del L747_A750), an *EGFR* activating mutation that makes patients eligible for EGFR inhibitor therapy, was detected in 63.2% of FTCs in one patient. In a second patient, the *EML4-ALK* (echinoderm microtubule associated protein-like 4–anaplastic lymphoma kinase) fusion variant, which is a novel target in a subset of non-small cell lung cancer cases, was detected in 85% of isolated FTCs. The *ALK G1202R* mutation, a known Alectinib-resistance mutation, was the only mutation identified throughout multiple FTC samples from another patient [13].

### 3.4. Colorectal Cancer

Colorectal cancer (CRC) is the third most commonly diagnosed cancer and second most common death-causing cancer in Australia. It is a lethal cancer with a high mortality rate due to distant metastasis. A number of driver genes are commonly identified in CRC, including mutated *BRAF*, *KRAS*, *EGFR* and *PIK3Ca* [19,76,84]. EGFR is the main therapeutic target; however, responses to EGFR inhibition are variable [19]. The key mutations found in single-cell analysis of CRC CTCs so far are *KRAS*, *PIK3CA* and *EGFR* mutations. Significant heterogeneous expression of *KRAS*, *PIK3CA* and *EGFR* was found among CTCs within the same patient and between different individuals [19,76]. A mutational discordance between primary tumor tissue and CTC WGAs was observed for *KRAS*, and remarkably different *KRAS* mutations in different single-CTC WGAs from the same individual patients have been observed [19,76]. CTCs were observed with increased EGFR expression in some patients, and *EGFR* gene amplification was identified in 7 out of 26 CTC WGAs for three patients [19].

### 3.5. Other Cancer Types

Pancreatic cancer is a lethal cancer with a less than 10% 5-year survival rate. *KRAS* is the predominant mutated gene in pancreatic cancer, and targeting KRAS may be an attractive therapy, despite many trial failures for anti-KRAS therapies [85]. *KRAS* mutations have been detected in 92% of patients, with a detection rate of 27.7% in total single-CTC WGAs (REPLI-g, MDA), but not in any WGAs of control WBCs. Interestingly, at least 10 single CTCs are required to reliably detect the *KRAS* heterozygous allele [77], which indicates that single-cell amplification bias responsible for ADO can be reduced by sequencing at least 10 cells together. In a study on single-CTC analysis of melanoma [79], *CDKN2A* and *PTEN* deletions and amplifications of *TERT*, *BRAF*, *KRAS* and *MDM2* were found. Moreover, new chromosomal amplifications of chromosomes 12, 17 and 19 were detected [79].

Studies on other cancer types are still rare, while there are also a few ongoing clinical trials recruiting patients (NCT05242237: liver cancer; NCT04568291: lung cancer with bone metastases). More studies are needed.

## 4. Concluding Remarks

The field of single-CTC analysis is still in its infancy. The power of single-cell WGA and downstream single-cell genetic profiling with individual CTC WGAs has been demonstrated in various cancer types over the past ten years. However, some emerging biomarkers, such as PDL-1 and other immune checkpoint markers, have not been indicated in any of the single-cell analysis studies so far. The main reason for this may be that large numbers of studies are performed with targeted gene panels which may not cover these markers, while the sample sizes of current studies are normally small, with limited access to CTCs or liquid biopsies. Although such markers are an interesting and hot topic, considering the proportion of positive cancer cells in a whole tumor, the chance of discovering these markers in CTCs will be relatively low. Furthermore, the intrinsic limitations of WGA (potential genome loss, etc.) might play roles in the attempts to find markers. Therefore, technique development and further studies with large sample numbers are warranted.

Despite the length of the procedure from a blood draw to the completion of genetic profiling, with emerging technical advances in WGA, single-CTC profiling will become more accurate and convenient and has already demonstrated strong potential to guide personalized therapy. In particular, the capability of detecting heterogeneity sets single-CTC analysis apart from biomarker detection using cell-free DNA or exosome analysis. To bring these analyses into diagnostic settings, it is desirable to develop standardized CTC isolation and WGA technologies that will allow data comparison worldwide.

## Figures and Tables

**Figure 1 ijms-23-08386-f001:**
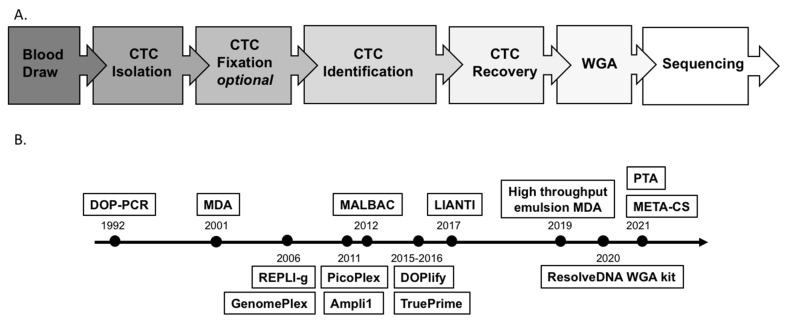
(**A**) Workflow of single-CTC analysis. (**B**) Timeline of WGA technology development. Above the timeline: key methodologies; below the timeline: availability of some commercial kits. Note: CTC: circulating tumor cell; DOP-PCR: degenerate oligonucleotide-primed PCR; MDA: multiple displacement amplification; MALBAC: multiple annealing and looping-based amplification cycles; LIANTI: linear amplification via transposon insertion; PTA: primary template-directed amplification; META-CS: multiplexed end-tagging amplification of complementary strands.

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
