# Peer review of "Single-Circulating Tumor Cell Whole Genome Amplification to Unravel Cancer Heterogeneity and Actionable Biomarkers"

_ijms, 2022, doi:10.3390/ijms23158386_

Round 1

Reviewer 1 Report

Dear Colleagues,

“Single Circulating Tumor Cell Whole Genome Amplification to unrvael Cancer Heterogeneity and Actionable Biomarkers” by dr Tanzila Khan et al. is an interesting and well-organized review of scientific evidence revolving aroung Whole Genome Sequencing and its applications, retrieving articles published in past 10 years.

The structure of the Manuscript is simple and easy to understand; I deeply appreciated the detailed explanation of every method cited; however, adding a timeline of method development (similar to the one nicely depicted in figure 1) would have been even more clarifying.

While the article itself revolves around an amazing and challenging part of Oncology and Bio-Oncology, there are some flaws that need to be addressed:

-       English: Several typos could be detected throughout the text, so an English revision is mandatory

-       Discussion and intrinsic limitations: a separate discussion section should be implemente ( for example, some important prognostic and predictive markers (such as PDL1) could not be detected through liquid biopsy and WGA at present days; articulating why, how and when to use and promote WGA and which intrinsic limitations are bound to be faced would improve the overall quality of the study itself). This suggestion could be rejected by authors if they properly debate.

-       Section 3.4 colorectal cancer: there is no epidemiology introduction; in consistency with other tumor specific sections, please re-elaborate.

-       Figures and Tables are well written, well presented, well structured.

Kind Regards

Author Response

Dear Colleagues,

“Single Circulating Tumor Cell Whole Genome Amplification to unrvael Cancer Heterogeneity and Actionable Biomarkers” by dr Tanzila Khan et al. is an interesting and well-organized review of scientific evidence revolving aroung Whole Genome Sequencing and its applications, retrieving articles published in past 10 years.

The structure of the Manuscript is simple and easy to understand; I deeply appreciated the detailed explanation of every method cited; however, adding a timeline of method development (similar to the one nicely depicted in figure 1) would have been even more clarifying.

We thank the reviewer for their overall positive comments and valuable suggestions. We added a sub-figure 1B to address the timeline of WGA strategy developments and amended the figure legend.

While the article itself revolves around an amazing and challenging part of Oncology and Bio-Oncology, there are some flaws that need to be addressed:

-       English: Several typos could be detected throughout the text, so an English revision is mandatory

Thank the reviewer for pointing this out! We have checked thoroughly and revised the manuscript for language issues (typos, capital letter, etc), and changed accordingly as highlighted in manuscript with tracked changes. 

Besides, we also double checked the cited references and updated the references 11,43,44,57,59 with complete formats. We also add a few extra references when we did changes as reviewer 1-3 suggested.

-       Discussion and intrinsic limitations: a separate discussion section should be implement ( for example, some important prognostic and predictive markers (such as PDL1) could not be detected through liquid biopsy and WGA at present days; articulating why, how and when to use and promote WGA and which intrinsic limitations are bound to be faced would improve the overall quality of the study itself). This suggestion could be rejected by authors if they properly debate.

      We appreciate reviewer’s comments. PDL-1 and other ICI markers have not been indicated in any of the single cell analysis studies so far. The main reason would be large amount of studies are performed with targeted gene panels which do not cover these markers yet and the sample sizes of current studies are normally small with limited access of CTC/liquid biopsy. Although PDL1 gene amplification is an interesting topic, which can be potentially measured with FISH on tissue samples or ddPCR with specific primers/probes on CTC, however considering the proportion of PDL1 positive cancers in whole tumor, the chance to find PDL1 amplification on CTC will be very low. Also intrinsic limitation of WGA (potential genome loss, etc) might also play roles in finding such emerging markers. Therefore, technique development and further studies with large sample numbers are warranted.

We added some lines in the discussion part (line 397-406) to address this question.

-       Section 3.4 colorectal cancer: there is no epidemiology introduction; in consistency with other tumor specific sections, please re-elaborate.

      This is a good point made by the reviewer and we added a small introductory section in CRC section.

-       Figures and Tables are well written, well presented, well structured.

Thanks for the appreciative comments.

Reviewer 2 Report

Revision of the manuscript “ijms-1816040” by Khan et al. The manuscript has an interesting proposal, but there are some issues to be addressed before acceptance, as follows:

1. Authors could exemplify and discuss the treatments mentioned on lines 45-52.

2. Section 2.1 could discuss in details the advantages and disadvantage among the mentioned methods. For instance, comparing the CTC isolation methods (CellSarch vs. CellCollector), CTC recovery methods (CellCollector vs. FACS), WGA kits, etc.

3. Authors could add information about ongoing clinical trials, and studies on bioRxiv.org or conferences, such as https://aacrjournals.org/cancerres/article/82/12_Supplement/1700/703123

Author Response

Reviewer 2:

Revision of the manuscript “ijms-1816040” by Khan et al. The manuscript has an interesting proposal, but there are some issues to be addressed before acceptance, as follows:

  1. Authors could exemplify and discuss the treatments mentioned on lines 45-52.

Thanks for the suggestion and we added text and reference in line 49

  1. Section 2.1 could discuss in details the advantages and disadvantage among the mentioned methods. For instance, comparing the CTC isolation methods (CellSarch vs. CellCollector), CTC recovery methods (CellCollector vs. FACS), WGA kits, etc.

Thanks for the valuable comments and we have added a few lines in line 89-91 to mention CellCollector. Other comparisons on WGA kits are listed in table 1 and text.

  1. Authors could add information about ongoing clinical trials, and studies on bioRxiv.org or conferences, such as https://aacrjournals.org/cancerres/article/82/12_Supplement/1700/703123

Thanks for this valuable suggestion! When we generated table 2, we elaborately listed all available studies and we apologise that maybe a few small studies (if any) which may lack of clear description of methods/results or with no significant findings are missing. In this review, we only include the studies with full text available, this is also why we did not include any poster and conference study with limited data. As reviewer suggested, we mentioned 2 ongoing clinical trial studies in line 391.

Reviewer 3 Report

The manuscript "Single Circulating Tumor Cell Whole Genome Amplification to  Unravel Cancer Heterogeneity and Actionable Biomarkers" is a review focused on SCTC whole genome amplification in cancer detection. The topic is interesting and actual. The manuscript is correctly written (minor errors or incorrect details are highlighted on the manuscript in attachment).

However, the manuscript has a major issue, which is a lack of details regarding the topic in distinct cancers. For example, colorectal cancer is covered by a very short paragraph. Some cancer types are not mentioned at all. A review should cover a systematic and comprehensive overview of the latest knowledge in the field. Therefore, the text should be enriched with data for different types of cancer as well as available data/published articles relative to experiences from relevant laboratories that use this technique.

Author Response

Reviewer 3:

The manuscript "Single Circulating Tumor Cell Whole Genome Amplification to Unravel Cancer Heterogeneity and Actionable Biomarkers" is a review focused on SCTC whole genome amplification in cancer detection. The topic is interesting and actual. The manuscript is correctly written (minor errors or incorrect details are highlighted on the manuscript in attachment).

However, the manuscript has a major issue, which is a lack of details regarding the topic in distinct cancers. For example, colorectal cancer is covered by a very short paragraph. Some cancer types are not mentioned at all. A review should cover a systematic and comprehensive overview of the latest knowledge in the field. Therefore, the text should be enriched with data for different types of cancer as well as available data/published articles relative to experiences from relevant laboratories that use this technique.

Thanks for the overall positive comments. We understand where the reviewer is coming from, however the aim of this manuscript is to review how whole genome amplification techniques have been applied in circulating tumour cells to identify biomarkers and heterogeneity. As such, we have performed a thorough publication search through PubMed for the recent decade. We elaborately listed all available studies in table 2 that analysed patient cohorts or CTC numbers to provide significant findings. We appreciate that we might have missed some relevant publications but believe we have covered the field well. It is the case that most studies focus on certain (common) cancers and do not cover some rare cancer types (for example, brain cancer, renal cancer, etc). The field is still emerging and more studies on other cancer types are urgently needed.

Nevertheless, it is almost certain that the findings and technical applications are transferable to other cancers when CTCs are isolated.

In regard to colorectal cancer, we have now added some more generic information regarding this cancer (line 367, also as requested by reviewer 1).  We also correct other minor errors as highlighted in manuscripts with tracked changes.